# Clinical Utility of Preoperative Assessment in Ovarian Cancer Cytoreduction

**DOI:** 10.3390/diagnostics10080568

**Published:** 2020-08-07

**Authors:** Pratistha Koirala, Ashley S Moon, Linus Chuang

**Affiliations:** Department of Obstetrics and Gynecology, Division of Gynecologic Oncology, Danbury Hospital, Danbury, CT 06810, USA; ashmoon@stanford.edu (A.S.M.); linus.chuang@wchn.org (L.C.)

**Keywords:** ovarian cancer cytoreduction, neoadjuvant chemotherapy, preoperative predictors of optimal tumor cytoreduction

## Abstract

Ovarian cancer is the deadliest gynecologic cancer, in part due to late presentation. Many women have vague early symptoms and present with disseminated disease. Cytoreductive surgery can be extensive, involving multiple organ systems. Novel therapies and recent clinical trials have provided evidence that, compared to primary cytoreduction, neoadjuvant chemotherapy has equivalent survival outcomes with less morbidity. There is increasing need for validated tools and mechanisms for clinicians to determine the optimal management of ovarian cancer patients.

## 1. Introduction

Although ovarian cancer represents only 1.3% of newly diagnosed cancers in the United States, it accounts for 2.3% of overall cancer related deaths [1]. In women, ovarian cancer is the second most common but deadliest gynecologic cancer, and the fourth deadliest overall cancer [2]. Approximately 80–90% of ovarian cancers occur in women between the ages of 20- and 65-years-old, and 60% of ovarian tumors are of epithelial origin [3]. Currently, the five-year-survival rate for ovarian cancer is less than 50%. Prophylactic salpingectomy can be performed opportunistically at the time of other surgeries, such as cesarean section or during non-gynecologic procedures, and has been shown to reduce ovarian cancer risk [4]. Due to a lack of effective screening methods and vague symptoms, ovarian cancer often presents at later, disseminated stages, leading to higher mortality compared to less prevalent cancers. With the exception of high risk women with known hereditary cancer gene mutations, the United States Preventive Services Task Force recommends against ovarian cancer screening in asymptomatic women as current methods have high false positive rates which can lead to unnecessary surgical interventions [5]. Due to the inability to detect ovarian cancer at early stages, over two-thirds of newly diagnosed cases present at stage III and IV [6].

Historically, ovarian cancer presenting at any stage was managed with primary cytoreductive surgery followed by adjuvant chemotherapy. Recent literature has demonstrated that advanced stage cancers have improved surgical and equivalent survival outcomes with neoadjuvant chemotherapy. Within the last few years, poly ADP ribose polymerase (PARP) inhibitors have demonstrated improved survival and reduced risk of recurrent disease in newly diagnosed advanced ovarian, fallopian tube and peritoneal cancer after response to first-line platinum neoadjuvant chemotherapy in homologous recombinant deficiency populations [7]. This was based on prior trials that showed PARP inhibitors increased progression-free survival in platinum-sensitive recurrent ovarian cancer in both breast cancer gene (BRCA) mutated and non-BRCA mutated patients [8]. These changes in management necessitate clinicians to utilize tools to stratify patients to the most appropriate management.

## 2. The Role of Neoadjuvant Therapy versus Primary Cytoreduction

Until the recent advancement of PARP inhibitors, the pillars of ovarian cancer treatment have been cytoreductive surgery and chemotherapy. Both techniques are utilized, but the sequence with which they are used is dependent on stage and presentation. In cases where optimal cytoreduction can be achieved, the preferred treatment method is primary surgery followed by a platinum-based chemotherapy regimen [9,10].

The Gynecologic Oncology Group (GOG) has previously defined optimal tumor cytoreduction as less than 1 cm of residual disease [11]. There is evidence that minimizing residual tumor to any degree improves survival outcomes, with one study demonstrating median survival ranging from 48 to 66 months depending on the degree of residual tumor (less than 1 cm remaining versus median survival of 106 months in patients with no residual disease after primary cytoreduction [12]). There is evidence that less than 1 cm of residual disease has efficacy, but the current goal of cytoreductive surgery is to achieve complete tumor resection with no evidence of remaining disease [11,13].

If there is concern that optimal cytoreduction cannot be achieved, patient outcomes are improved by administering neoadjuvant chemotherapy followed by interval tumor cytoreduction. In the last decade, this paradigm shift towards neoadjuvant treatment prior to cytoreductive surgery has maintained survival outcomes with less surgical morbidity.

The 2010 European Organization for Research and Treatment of Cancer (EORTC) trial was the first major study to address the outcomes between primary cytoreduction versus neoadjuvant chemotherapy. In this study, 632 patients with either stage IIIC or IV ovarian, fallopian tube, or primary peritoneal carcinoma were randomly assigned to either undergo platinum-based neoadjuvant chemotherapy, followed by interval or primary cytoreductive surgery, followed by adjuvant platinum-based chemotherapy. For patients undergoing neoadjuvant chemotherapy, the hazard ratio was 0.98 for death and 1.01 for progressive disease, showing that this treatment was non-inferior to the standard of care at that time. Postoperative complications and mortalities were higher after primary cytoreduction [14].

The 2015 primary chemotherapy versus primary surgery for newly diagnosed advanced ovarian cancer (CHORUS) trial performed similar randomization of 550 patients with stage III or IV ovarian cancer. Their findings replicated those of the EORTC trial, with a hazard ratio of 0.87 favoring primary chemotherapy in their patient population [15]. A 2016 Japan Oncology Group Study confirmed these findings in a randomized control trial of 301 patients. Not only did they find neoadjuvant chemotherapy non-inferior, they also noted that these patients ultimately underwent less invasive surgical procedures and suggested that this may be a new standard of care [16].

A 2016 superiority trial, primary surgery versus neoadjuvant chemotherapy in advanced epithelial ovarian cancer with high tumor load (SCORPION trial), randomized 110 women with ovarian cancer to either undergo primary cytoreduction or neoadjuvant chemotherapy. In this trial, rather than using stage to qualify a cancer as ‘advanced’, all participants underwent diagnostic laparoscopy to determine tumor burden and the extent of their disease. This trial demonstrated that not only was neoadjuvant chemotherapy superior in mortality outcomes, but that it was associated with improved quality of life metrics compared to primary cytoreductive surgery [17].

Ultimately, complete surgical resection of ovarian cancer remains the standard of care, but there are multiple approaches to achieving this goal. The ability to achieve this goal is multifactorial, ranging from the properties of each individual tumor to the ability of the surgeon performing cytoreduction. Many early stage ovarian cancers can be managed with primary cytoreduction with the addition of adjuvant chemotherapy, whereas stage IV ovarian cancers have a variable chance of optimal primary cytoreduction (ranging from 15% to 85% [18,19]). Poor functional and nutritional status may deem a patient an unsatisfactory candidate for primary cytoreduction. In these cases, neoadjuvant therapy can be started and the patient is reassessed (often at the midpoint of a six-cycle regimen). If appropriate, interval cytoreduction can be performed at that time.

The branch point in determining if a patient is a candidate for primary versus interval cytoreductive surgery is ill-defined. There are currently no high-fidelity models to predict success rates in achieving optimal cytoreduction, leading to suboptimal cytoreduction and large residual disease burden in some patients. In the aforementioned studies, the included eligible patients were those with biopsy-proven stage IIIC or IV primary ovarian, peritoneal, or fallopian-tube carcinoma, the presence of a pelvic mass, the presence of at least a 2 cm metastasis outside the pelvis, or regional lymph node metastasis. A ratio of serum cancer antigen-125 (CA-125) to carcinoembryonic antigen (CEA) of 25 or higher was required along with other negative metastatic evaluations including mammography and gastroenterology studies. Patients had to have a World Health Organization performance status of 0 (asymptomatic) to 2 (symptomatic) and have no serious disabling diseases that precluded them from undergoing cytoreductive surgeries [14,15,16,17].

Efforts have been made to standardize preoperative assessment in determining surgical versus medical management. Nomograms have been developed to predict disease-specific survival for advanced ovarian cancer patients who underwent cytoreductive surgery and adjuvant platinum-based therapy [20,21,22]. Prognostic factors were studied such as age, performance status, tumor histology, residual disease. External validation studies of these predictive models have been performed, which showed adequate applicability, but concluded that missing data and a wide range of variables diminished their clinical utility [23]. In this review, we discuss some of these clinical predictors, including tumor and molecular markers, preoperative imaging, functional status, and diagnostic laparoscopy. We further comment on clinical decision making and management in resource limited settings.

## 3. Functional Status

Nearly half of ovarian cancer patients are over the age of 64 and 25% are over the age of 74 [1]. Many patients present at late stages with multiorgan involvement and manifestations such as ascites. In addition to disease related impairments on functional status, many older patients have comorbid conditions. Regardless of comorbidities, age is an independent prognostic factor, with patients over 80 having significantly higher perioperative mortality and morbidity, including kidney failure, pulmonary, and cardiovascular compromise [24,25]. Perioperative complication rates increase with each age group and are further compounded by tumor burden [26]. In another study, although age was not found to be prognostic, older patients consistently had worse outcomes despite intention to treat [27]. Due to the association of increasing age with poorer outcomes, we recommend to consider a pre-operative functional and nutritional status analysis in patients over 65 years of age.

Multiple studies found that low serum albumin levels, a stand in for poor nutritional status, were negatively associated with survival and that this association was durable across all stages of ovarian cancer and other history [25,26,28]. In one study, subgroup analysis demonstrated significantly poorer survival in laparotomy but not laparoscopic cases. Hypoalbuminemia is associated with increased resource utilization, such as longer hospital and intensive care unit length of stay [29]. In gastrointestinal surgeries, hypoalbuminemia was associated with increased rates of surgical site infections. Serum albumin levels provide a noninvasive method to assess the risk associated with surgical intervention [30]. Specifically, a preoperative albumin of less than 3.5 g/dL has been associated with poor survival outcomes in multiple studies [28,29,30]. A meta-analysis by Ge et al. found that 0.1 g/dL increases in serum albumin levels were significantly associated with improved survival outcomes [31]. Patient with severe hypoalbuminemia should be considered for nutritional support for one to two weeks to optimize surgical outcomes [32,33].

Elderly patients often receive less aggressive treatment. In one study, elderly patients with poor nutritional status and extensive tumor burden had increasing morbidity with no added survival benefit [34]. Patients often also have delays in the initiation of adjuvant chemotherapy or reductions in dosage due to complications of cytoreductive surgery. These delays are more likely to occur in elderly patients and are associated with significantly worse overall survival [35]. Despite potentially worse outcomes in elderly patients, surgical intervention should still be considered for otherwise appropriate candidates being managed in high volume centers with multidisciplinary care teams [27,36]. In addition to age and nutritional status, as measured by serum albumin levels, a patient’s cardiopulmonary function and status can be evaluated using the American Society of Anesthesiologists Physical Classification System (ASA score) which accounts for comorbid conditions [37]. Patients with higher ASA scores tend to have worse postsurgical outcomes [24,38]. These parameters can play a role in creating individualized treatment plans and care should be taken to optimize nutritional and functional status prior to cytoreductive surgery or chemotherapy.

## 4. Tumor Markers

Tumor and molecular markers are widely used and serve a wide range of functions in cancer screening, detection and diagnosis, treatment planning, and disease monitoring. Screening tests must have high sensitivity and remain cost effective for maximal impact while diagnostic tests must have high specificity [39]. Currently, ovarian cancer tumor markers play a role in post-treatment disease monitoring but have a limited role in screening or diagnosis due to their lack of specificity. CA-125 is the most studied tumor marker for ovarian cancer. However, CA-125 can be elevated in non-gynecologic disease processes such as hepatitis, and in benign gynecologic conditions including normal menses [40]. Up to 50% of ovarian cancer cases can have normal CA-125 levels. Despite its limitations, CA-125 is currently the standard biomarker in ovarian cancer management and is used to monitor recurrence and disease response to treatment [41,42].

Three major studies have been performed over the past two decades to analyze the utility of CA-125 in combination with transvaginal ultrasound: as part of the Prostate, Lung, Colorectal, and Ovarian (PLCO) Cancer Screening Randomized Control Trial, the UK Collaborative Trial of Ovarian Cancer Screening (UKCTOCS), and the Shizuoka Cohort Study of Ovarian Cancer Screening (SCSOCS) trial [41,43,44]. The PLCO and SCSOCS trials demonstrated no significant difference in ovarian cancer mortality between women who had screening and those who did not [41,45]. Conversely, screening of asymptomatic women led to increased rates of surgical interventions and serious complications [45]. On the other hand, in a post-hoc analysis, the UKCTOCS showed long term decreased ovarian cancer mortality when using CA-125 in conjunction with the Risk of Ovarian Cancer Algorithm (ROCA) [46]. Due to the cost effectiveness of screening and a lack of clear evidence-based guidelines in the United States, CA-125 is not commonly used as a screening tool [47,48,49].

CA-125 may play a role in guiding primary cytoreduction versus neoadjuvant chemotherapy. In a retrospective study, Chi et al. found that preoperative serum CA-125 was correlated with successful optimal cytoreduction. The authors found that a preoperative CA-125 cut-off of 500 U/mL was highly predictive of optimal cytoreduction, with values over the threshold resulting in suboptimal cytoreduction 80% of the time [50]. A second study evaluated this cut-off in their patients and reported a lower predictive value for a CA-125 of 500 U/mL, but noted that no patients with a CA-125 over 1500 U/mL had optimal cytoreduction [51]. The authors suggested utilizing diagnostic laparoscopy to make a final decision for patients who had a CA-125 value greater than their cut-off of 500 U/mL [50].

Recent data have suggested that CA-125 and human epididymis protein 4 (HE4) can be used to stratify treatment algorithms. HE4 was identified as a gene of interest in ovarian cancer and has shown promise in differentiating benign versus malignant ovarian lesions. Similar to CA-125, HE4 levels fluctuate in relation to disease burden [52]. When multiple markers were studied, including CA-125, only HE4 had prognostic value immediately after surgical cytoreduction [53]. Other studies have shown that a combination of HE4 and CA-125 may have a higher diagnostic utility and surgical outcome predictability than either marker alone [54,55]. Cutoff values of preoperative serum HE4 and CA-125 of 777.10 pmol/L and 313.60 U/mL, respectively, were suggested by the authors to accurately predict outcomes of cytoreductive surgeries [55]. A study by Chudecka-Głaz found that HE4 may be an independent prognostic factor of chemotherapeutic treatment response. In this study, the normalization of HE4 after therapy or a 50% reduction of HE4 levels before interval surgery was associated with a significant improvement in both progression-free and overall survival [56]. Although these studies are promising, the results are still preliminary, based on retrospective studies, and sample sizes were small. Larger studies are needed for tumor markers to play a standard role in the stratification of surgical and medical management.

## 5. Preoperative Imaging

Transvaginal ultrasound (TVUS) is the mainstay in evaluating adnexal masses and is often the modality of choice given its clinical utility and cost effectiveness. TVUS is both sensitive and specific in the initial differentiation of benign versus malignant masses; when combined with color flow doppler, pattern recognition, and clinical background, TVUS reaches a sensitivity of 85% and a specificity of 90% [57,58]. In some institutions, such as referral centers with expert operators, ultrasound has displaced computed tomography (CT) in the pre-operative evaluation of pelvic and abdominal masses [59]. Ultrasound alone has not been used to predict successful cytoreduction although it has been used as part of larger predictive models such as ROCA, which combines serum biomarkers and TVUS. Compared to ROCA, ultrasound alone has decreased positive predictive value [60]. A study is currently underway to further characterize the role of transvaginal ultrasound in predicting successful cytoreduction in ovarian cancer [61].

CT is an important tool in ovarian cancer diagnosis and staging. Its primary role is determining the extent of disease. On the other hand, the National Comprehensive Cancer Network and Society for Gynecologic Oncology guidelines recommend CT scans only when clinically indicated [62,63]. Choosing Wisely^®^, a campaign on best practices from the American Board of Internal Medicine and the National Physician Alliance, advises against its routine use for surveillance [64]. Esselen et al. found that despite these guidelines, CT is commonly used in ovarian cancer surveillance even in the absence of clinical indications [65].

Multiple studies have attempted to validate CT as a predictor for successful primary surgical cytoreduction, with conflicting results. Some studies have suggested that CT may be a valuable predictor and specific findings, including mesenteric involvement and high grade ascites, may decrease the success of surgical cytoreduction [66,67,68,69]. Conversely, other studies found that CT was not predictive, or that it was equivocal, and cautioned against utilizing CT due to limited data and the small sample sizes of previous works [70,71]. A meta-analysis of these studies found that CT alone is a poor predictor of optimal cytoreduction and should be used with caution [72].

CT may have more utility in conjunction with other screening tools. Bristow et al. described a Predictive Index Score based on multiple radiographic findings, CA-125 levels, and functional status [73]. Subsequent studies have found that a combination of preoperative platelet count, functional status, and diffuse peritoneal thickening or ascites on CT was associated with suboptimal cytoreduction, but noted that the results needed external validation [74,75]. Multiple small trials have also assessed the predictive value of positron emission tomography (PET). These early results suggest a potential role for PET/CT in predicting optimal primary or secondary cytoreduction. Overall, CT continues to be widely used although evidence is contradictory and future large-scale studies are warranted.

Magnetic resonance imaging (MRI) is not commonly used in the management of ovarian cancer but it plays a role in characterizing difficult-to-assess adnexal masses. To date, there are limited data on the role that MRI can play in ovarian cancer management. One study used MRI to determine the Peritoneal Cancer Index (PCI), a scoring system based on sites of disease in the abdomen. This study found that MRI was useful in determining the PCI and that it may have a role in predicting successful cytoreduction, but this study was limited with only 25 participants [76]. A second study had similar findings in a slightly larger sample of 50 patients, but further research is needed prior to expanding the role of MRI in the diagnosis and management of ovarian cancer [77].

## 6. Minimally Invasive Approaches and Diagnostic Laparoscopy

Ovarian cancer cytoreduction surgery has traditionally been performed by laparotomy with total abdominal hysterectomy, bilateral salpingo-oophorectomy, omentectomy, peritoneal biopsies, pelvic and paraaortic lymph node biopsy versus lymphadenectomy, and resection of all visible disease, which can involve other organ systems such as the bowel, spleen, and liver. Multiple studies have demonstrated that minimally invasive techniques have value in evaluation, diagnosis, staging, and management of ovarian cancer [78]. To date, published studies have been retrospective analyses of small, carefully selected cohorts [79,80]. Both laparoscopic and robotic approaches have been utilized [81]. Despite this preliminary evidence, further research is needed on the role of minimally invasive cytoreductive surgery in ovarian cancer [82].

One strength of diagnostic laparoscopy is that initial evaluation can be performed, allowing the surgeon to directly visualize the tumor burden prior to making a decision on how to proceed. A 2005 study by Fagotti et al. sought to determine the accuracy of laparoscopic assessment in a randomized clinical trial of 95 patients, 64 of whom met the inclusion criteria. Patients who met the inclusion criteria underwent diagnostic laparoscopy, at which point the surgeon determined if the patient was a candidate for primary cytoreduction or if she were better suited for neoadjuvant chemotherapy. Regardless of the evaluation, all patients then underwent laparotomy and primary cytoreduction. In this powerful study, although some patients who were deemed appropriate for primary surgery had suboptimal cytoreduction, there were no patients who had been deemed unsuitable for cytoreductive surgery whose tumors were, in fact, optimally debulked [83]. Multiple studies have utilized the model proposed by Fagotti et al. and demonstrated its utility in their patient populations and for interval cytoreduction after neoadjuvant chemotherapy [84,85,86]. A Cochrane review of diagnostic laparoscopy suggests that it may have a role in the management of ovarian cancer, but the data are limited [87]. To date, these studies show that laparoscopy has the potential to avoid ‘futile laparotomies’ [88].

## 7. Liquid Biopsy

Liquid biopsies, most commonly acquired through blood samples, allow for tumor surveillance and have the potential to tailor treatment for precision medicine. Liquid biopsy utilizes circulating tumor cells (CTCs), circulating tumor DNA (ctDNA), circulating cell-free microRNAs, and circulating exosomes to diagnose and monitor cancer. CTCs are more commonly found in advanced stage (III and IV) than early stage ovarian cancer [89]. They are released from primary, metastatic or recurrent ovarian cancers and can be identified in the peripheral blood [90]. It is technically challenging to detect CTCs in whole blood as the ratio of CTCs to non-tumor cells is low. Immunocytochemistry or gene expression analysis such as real-time PCR (RT-PCR) allow for the enhanced detection of CTCs [91,92]. Potential biomarkers on CTCs have been uncovered using molecular profiling [93].

CTCs have been shown to be a predictor of treatment response [92]. Patients with platinum-resistant ovarian cancer have substantially higher CTCs when compared to those with platinum-sensitive tumors [94]. ctDNA may be more reliable that CTCs as a biomarker of response to treatment and long-term prognosis. *TP53* mutant-ctDNA detection rates of 75% to 100% have been reported in patients with high grade serous ovarian cancer [94,95]. Persistence of *TP53* gene variants in ctDNA after neoadjuvant chemotherapy correlated to the presence of minimal residual disease [96]. ctDNA in ovarian cancer has been shown to be more accurate than CA-125 and imaging to predict tumor responses [97].

MicroRNAs (miRNAs) are a class of noncoding RNA with a length of 20–25 nucleotides. As a result of the deregulation of miRNAs, changes in the target genes lead to ovarian cancer progression [98]. Plasma miRNA levels may serve as a diagnostic biomarker for ovarian cancer [98,99]. Exosomes are 30–150 nm-sized extracellular vesicles of endocytic origin which play key roles in cancer biology by mediating cell-to-cell communication through the transfer of proteins, nucleic acids, and lipids [100]. Exosome research has rapidly expanded over the last decade in ovarian cancer. Recent studies have demonstrated that exosomes promote peritoneal dissemination through an interaction between cancer cells and their microenvironments. Targeting exosomes may have potential as diagnostic and therapeutic biomarkers for ovarian cancer [97,100]. Liquid biopsy to detect tumor cells and monitor aberrant gene expression are promising in their ability to detect, diagnose, and monitor ovarian cancer. With further studies, they may also represent tools to triage and predict treatment response, such as neoadjuvant chemotherapy versus primary cytoreduction.

## 8. Considerations in Resource Limited Settings

Gynecologic cancers are currently disproportionately represented in low resource settings with low and low/middle income countries (LIC and LMICs) expecting to see the largest growth in these malignancies [101]. There is evidence that patients with gynecologic malignancies have improved morbidity and mortality outcomes when treatment is driven by a trained gynecologic oncologist compared to a general obstetrician gynecologist or a surgical oncologist [102]. Many of these LIC and LMICs have variable or no training programs in gynecologic oncology and their ability to deliver care varies greatly, leading to global inequities in treatment [103,104]. Over the past decade, multiple initiatives have fostered the training of gynecologic oncologists globally with a focus on delivering surgical care [101].

In settings where both surgical and medical management are available, cost often limits the treatment choice. In Kenya where both modalities are available, chemotherapy is often unaffordable for many patients [105]. Multiple studies have evaluated the cost effectiveness of primary cytoreduction versus neoadjuvant chemotherapy. Three studies found that with the exception of high risk subgroups, such as patients older than 65 years, primary cytoreduction is more cost effective than neoadjuvant chemotherapy [106,107,108]. Although cost should not be the sole determinant of management, it will undoubtedly play a role in LIC and LMICs, where sometimes only surgical management is available. Conversely, at times of resource limitations and decreased surgical availability, even in higher income countries, neoadjuvant chemotherapy may become more prevalent. With a global pandemic and cancellation of non-urgent or emergent surgeries in most of the United States, it will be interesting to see the impact on management choices and long-term morbidity and mortality outcomes in these patients.

## 9. Summary

Improved surgical outcomes reported by the EORTC and the CHORUS trials have driven neoadjuvant chemotherapy as the initial treatment of choice for advanced stage ovarian cancers [15,109]. These trials demonstrated that neoadjuvant chemotherapy is non-inferior to primary cytoreduction for patients with stage IIIC and IV ovarian cancer. There has been controversy within the medical community on how to best apply these findings, with some groups arguing that all advanced ovarian cancer patients should undergo neoadjuvant chemotherapy while others maintain that primary cytoreduction has a role in well selected patients [9]. These factors were reviewed in this paper and their current utility in clinical decision making is summarized in Table 1. There have been multiple models that account for prognostic factors: functional status, tumor markers, imaging, and diagnostic laparoscopy in order to predict which patients are candidates for optimal cytoreduction. In Figure 1 we present a general algorithm for the evaluation and management of adnexal masses in a nonpregnant patient. Many of these trials are limited in their sample size, or have a wide range of variables, or unavailable data. Further studies, including genetic and molecular profiling, are needed to optimize a predictive model to help clinicians determine the best treatment course for their patients.

Evaluation of a patient with an adnexal mass should begin with a thorough history and physical exam. Once a suspicious mass has been identified, initial imaging with transvaginal ultrasound (TVUS) in conjunction with tumor markers is a powerful tool to differentiate likely benign from suspected malignant conditions, while a CT scan can assess for metastasis and extent of disease. In patients with an isolated mass, surgical intervention is warranted both for management and to provide tissue diagnosis. For patients with extensive disease on preoperative imaging, diagnostic laparoscopy allows the surgeon to assess the tumor under direct visualization and determine respectability. In cases of both an isolated or extensive tumor, functional status should be taken into account. All patients over the age of 65 years should undergo thorough nutritional and functional status screening to determine if they are good surgical candidates. Patients with serum albumin levels of less than 3.5 g/dL should be offered preoperative nutritional support to optimize surgical outcomes. For all patients, co-morbid conditions and cardiopulmonary status should be determined using a scale such as the American Society of Anesthesiologists Physical Classification System. Primary cytoreduction should be offered for patients who are good surgical candidates and whose tumor can be debulked with no visible tumor remaining. Neoadjuvant chemotherapy should be offered to patients who do not meet these criteria with interval reassessment for surgical candidacy. 

## Figures and Tables

**Figure 1 diagnostics-10-00568-f001:**
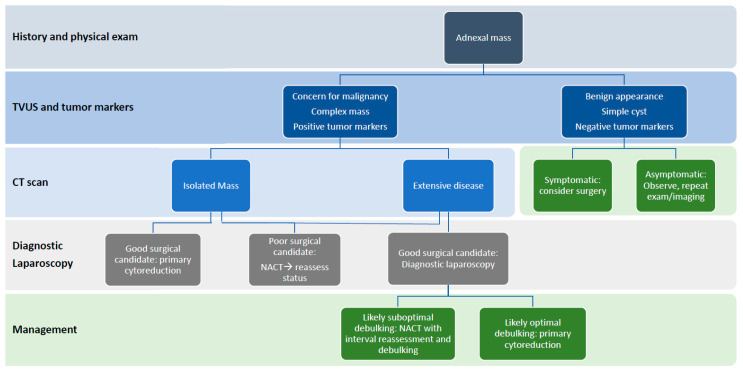
Preoperative assessment and management of an adnexal mass in a nonpregnant patient.

**Table 1 diagnostics-10-00568-t001:** Preoperative assessment. CT: computed tomography. MRI: magnetic resonance imaging.

Factors	Predictive Value
Age (≥65 years old)	High
Functional status (<4 metabolic equivalents)	High
Nutritional status (weight loss >5% premorbid weight)	High
Albumin level (≤3.5 g/dL)	High
Tumor markers (>500 units/mL)	Moderate
CT scan (ascites, carcinomatosis)	Moderate
MRI (carcinomatosis and liver metastasis)	Low to moderate
Diagnostic laparoscopy (carcinomatosis and extensive mesenteric involvement)	Moderate to high
Genetic and molecular markers	In research

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
