# Peer review of "Clinical Utility of Preoperative Assessment in Ovarian Cancer Cytoreduction"

_diagnostics, 2020, doi:10.3390/diagnostics10080568_

Round 1

Reviewer 1 Report

Good review of factors that can be used to assess the feasibility of an optimal cytoreduction in advanced ovarian cancer to help guide the clinician in upfront management (NACT vs PDS).  The conclusions are not novel but highlight the inexact nature of this decision.  This should be of moderate interest to readers and I have no major concerns regarding publication as written. 

Author Response

Reviewer 1

Good review of factors that can be used to assess the feasibility of an optimal cytoreduction in advanced ovarian cancer to help guide the clinician in upfront management (NACT vs PDS).  The conclusions are not novel but highlight the inexact nature of this decision.  This should be of moderate interest to readers and I have no major concerns regarding publication as written. 

Author response: We appreciate this feedback.

Reviewer 2 Report

Dear authors,

thank you very much for the helpful review.

I would like to suggest considering the addition of a brief discussion on the current state and potential value of liquid biopsy (also in combined use with traditional protein biomarkers) in the management of ovarian cancer.

Author Response

Reviewer 2

I would like to suggest considering the addition of a brief discussion on the current state and potential value of liquid biopsy (also in combined use with traditional protein biomarkers) in the management of ovarian cancer.

Author response: A new section, entitled ‘Liquid Biopsy’ was added on lines 259 to 289. The section gives an overview of multiple methods of liquid biopsy, including circulating tumor cells (CTCs), circulating tumor DNA (ctDNA), circulating cell-free microRNAs, and circulating exosomes.

This manuscript is a resubmission of an earlier submission. The following is a list of the peer review reports and author responses from that submission.

Round 1

Reviewer 1 Report

TITLE: Clinical utility of preoperative assessment in ovarian cancer cytoreduction.

Ovarian Cancer is an important topic in gynecological and oncological environment, which has a crucial impact on women’s health due to its high mortality.

We are increasingly looking for a tailoring of the treatment based on the best knowledge in molecular biology and molecular genetic fields (2), as well as on the patient's clinical conditions. A comprehensive preoperative assessment should consider these recent aspects, in order to identify early which tumors are responsive to chemotherapy and which one could be resistant.

However, this manuscript is only a narrative review, as a specific methodology approved for systematic reviews is not performed, such as the use of the PRISMA criteria (as also indicated in the instructions for the authors) which consists of use of the search strings, the presence of a second reviewer and of a flowchart.

Being a narrative revision, this paper result in a digression that does not have the ability to distinguish evidence from opinions.

Moreover a similar paper was masterfully draw up by Stephanie Lheureux, Charlie Gourley, Ignace Vergote, Amit M Oza, and published on the Lancet journal in 2019 (3).

Therefore, this paper does not provide significant news about ovarian cancer.

Specific comments:

Introduction

Line 18: 80-90% of all ovarian cancers occur in women between the ages of 20 and 65 and 60% of ovarian tumors are epithelial ones (1).

Line 25: Several studies propose an opportunistic prophylactic salpingectomy, during other surgery, in postmenopausal or premenopausal women who have exhausted their reproductive desire. (4)

Line 172: Moreover in a study published by Chudecka-GÅ‚az the effect of the normalization of the HE4 marker after therapy and 50% reduction of HE4 levels before interval cytoreductive surgery on PFS and OS was significant and that HE4 might be an independent prognostic factor of treatment response.(5)

The role of neoadjuvant therapy versus primary cytoreduction

Line 42: It would have been appropriate to indicate the specific reference (6)

Preoperative Images

Line 182: Even if ultrasound is a easy to use and diffuse in clinical practice, further data are  needed for the evaluation of the extent of disease in ovarian cancer. However, ultrasound has a high accuracy in staging advanced ovarian cancer patients. In a few cases, even if in referent centers and with expert operators, ultrasound has displaced CT in the pre-operative evaluation of pelvic and abdominal disease (7).

References:

  1. AIOM ovarian cancer guidelines 2019;
  2. Am J Pathol. 2016 Apr;186(4):733-47. doi: 10.1016/j.ajpath.2015.11.011. The Dualistic Model of Ovarian Carcinogenesis: Revisited, Revised, and Expanded. Kurman RJ1, Shih IeM2
  3. Stephanie Lheureux, Charlie Gourley, Ignace Vergote, Amit M Oza. Epithelial ovarian cancer. Lancet. 2019 Mar 23;393(10177):1240-1253;
  4. ACOG Committee Opinion No. 774: Opportunistic Salpingectomy as a Strategy for Epithelial Ovarian Cancer Prevention. Obstet Gynecol. 2019 Apr;133(4):e279-e284.
  5. Chudecka-Głaz A, Cymbaluk-Płoska A, Wężowska M, Menkiszak J. Could HE4 level measurements during first-line chemotherapy predict response to treatment among ovarian cancer patients? PLoS One. 2018 Mar 27;13(3):e0194270;
  6. Elattar A, Bryant A, Winter-Roach BA, Hatem M, Naik R. Optimal primary surgical treatment for advanced epithelial ovarian cancer. Cochrane Database Syst Rev. 2011 Aug 10(8):CD007565;
  7. De Blasis I, Moruzzi MC, Moro F, Mascilini F, Cianci S, Gueli Alletti S, Turco LC, Garganese G, Scambia G, Testa AC. Role of ultrasound in advanced peritoneal malignancies. Minerva Med. 2019 Aug;110(4):292-300;

Author Response

Author comment: The authors acknowledge that this was intended to be a narrative overview of the current status of ovarian cancer cytoreduction and neoadjuvant chemotherapy. This is not a systematic review and therefore formal criteria, such as  PRISMA, were not used. Papers were identified by performing a MEDLINE and google search as well as reviewing the bibliographies of relevant articles.

Specific comments:

Introduction

Line 18: 80-90% of all ovarian cancers occur in women between the ages of 20 and 65 and 60% of ovarian tumors are epithelial ones (1).

Author response: The following sentence was added to Line 18: “80-90% of ovarian cancers occur in women between the ages of 20 and 65-years-old and 60% of ovarian tumors are of epithelial origin”. The following reference was added to the manuscript:

Devouassoux-Shisheboran, M.; Genestie, C. Pathobiology of Ovarian Carcinomas. Chinese Journal of Cancer. Landes Bioscience 2015, pp 50–55. https://doi.org/10.5732/cjc.014.10273.

Line 25: Several studies propose an opportunistic prophylactic salpingectomy, during other surgery, in postmenopausal or premenopausal women who have exhausted their reproductive desire. (4)

Author response: “Prophylactic salpingectomy can be performed opportunistically at the time of other surgeries such as cesarean section or even during non-gynecologic procedures and has been shown to reduce ovarian cancer risk” was added to line 19 and the suggested reference was added to the manuscript.

Line 172: Moreover in a study published by Chudecka-GÅ‚az the effect of the normalization of the HE4 marker after therapy and 50% reduction of HE4 levels before interval cytoreductive surgery on PFS and OS was significant and that HE4 might be an independent prognostic factor of treatment response.(5)

Author response: “A study by Chudecka-GÅ‚az found that HE4 may be an independent prognostic factor of chemotherapeutic treatment response. In this study the normalization of HE4 after therapy or a 50% reduction of HE4 levels before interval surgery was associated with a significant improvement in both progression-free and overall survival” was added to line 169 and the suggested reference was added to the manuscript.

The role of neoadjuvant therapy versus primary cytoreduction

Line 42: It would have been appropriate to indicate the specific reference (6)

Author response: The suggested reference was added to the manuscript.

Preoperative Images

Line 182: Even if ultrasound is a easy to use and diffuse in clinical practice, further data are needed for the evaluation of the extent of disease in ovarian cancer. However, ultrasound has a high accuracy in staging advanced ovarian cancer patients. In a few cases, even if in referent centers and with expert operators, ultrasound has displaced CT in the pre-operative evaluation of pelvic and abdominal disease (7).

Author response: “Ultrasound is utilized in many clinical practices due to its ease of use and it is highly accurate in staging advanced ovarian cancer” was added to line 175 and “In some institutions, such as referent centers with expert operators, ultrasound has displaced Computer tomography (CT) in the pre-operative evaluation of pelvic and abdominal masses” was added to line 178. The suggested reference was added to the manuscript.

Reviewer 2 Report

This article is a descent review of the controversies surrounding the decision to proceed with either neoadjuvant chemotherapy (NACT) followed by interval debulking (IDS) versus primary cytoreduction (CRS) in ovarian cancer.

There are some parts that can probably be excluded based on the premise of the article (i.e. the section on tumor marker screening and the section on transvaginal US assessment of adnexal masses). 

It would be nice to have the authors propose an algorithm for NACT/IDS vs primary CRS that they recommend based on the factors listed in the article. 

Author Response

The sections on tumor markers and transvaginal ultrasound assessment were left in as they are both a part of the current standard preoperative evaluation of ovarian cancer.

An algorithm was created in graphical format and has been added as a figure to the manuscript.

Round 2

Reviewer 1 Report

We have read with interest the new version of the file, and we acknowledge the authors have modified the text according to reviewers suggestions.